# Recombination, meiotic expression and human codon usage

**Fanny Pouyet[1], Dominique Mouchiroud[1], Laurent Duret[1]\*, Marie Sémon[2]\***

[1]Laboratoire de Biométrie et Biologie Evolutive, Université de Lyon, Université Claude Bernard, Villeurbanne, France; [2]Laboratory of Biology and Modelling of the Cell, UnivLyon, ENS de Lyon, Univ Claude Bernard, CNRS UMR 5239, INSERM U1210, Laboratoire de Biologie et Modélisation de la Cellule, Lyon, France

**Abstract** Synonymous codon usage (SCU) varies widely among human genes. In particular, genes involved in different functional categories display a distinct codon usage, which was interpreted as evidence that SCU is adaptively constrained to optimize translation efficiency in distinct cellular states. We demonstrate here that SCU is not driven by constraints on tRNA abundance, but by large-scale variation in GC-content, caused by meiotic recombination, via the non-adaptive process of GC-biased gene conversion (gBGC). Expression in meiotic cells is associated with a strong decrease in recombination within genes. Differences in SCU among functional categories reflect differences in levels of meiotic transcription, which is linked to variation in recombination and therefore in gBGC. Overall, the gBGC model explains 70% of the variance in SCU among genes. We argue that the strong heterogeneity of SCU induced by gBGC in mammalian genomes precludes any optimization of the tRNA pool to the demand in codon usage.

DOI: https://doi.org/10.7554/eLife.27344.001

## Introduction

In humans, the usage of synonymous codons varies substantially among genes. Both adaptive and nonadaptive processes, not mutually exclusive, have been proposed to explain the existence of codon usage biases (*Duret, 2002*; *Chamary et al., 2006*; *Plotkin and Kudla, 2011*). The main adaptive model, called translational selection, proposes that synonymous codon usage (SCU) and abundance of tRNA are co-adapted to optimize the efficiency of translation (*Ikemura, 1981*; *Kanaya et al., 2001*; *Drummond and Wilke, 2008*; *Hershberg and Petrov, 2008*; *dos Reis and Wernisch, 2009*). The selective pressure on translational efficiency (in terms of both speed and accuracy) is expected to be more pronounced in highly expressed genes because they mobilize a large number of ribosomes (*Bulmer, 1991*) and are subject to stronger constraints on translational errors (*Akashi, 1994*; *Drummond and Wilke, 2008*). A first prediction of this model is that preferred codons should correspond to the most abundant tRNAs, particularly in highly expressed genes. A second prediction is that codon usage bias should correlate with gene expression patterns and tRNA contents. Both predictions are verified in some animals, such as flies and nematodes, the genomes of which show clear signatures of translational selection (*Shields et al., 1988*; *Duret and Mouchiroud, 1999*; *Duret, 2002*; *Castillo-Davis and Hartl, 2002*).

The situation is different in mammals, and notably humans, where the influence of translational selection is still strongly debated (*Duret, 2002*; *Chamary et al., 2006*; *Plotkin and Kudla, 2011*). It has long been shown that variation in SCU between genes is correlated to large-scale fluctuations of GC-content along chromosomes, the so-called isochores (*Bernardi et al., 1985*; *Mouchiroud et al., 1988*; *Mouchiroud et al., 1991*; *Clay and Bernardi, 2011*). The fact that codon usage correlates with the base composition of non-coding regions demonstrates that SCU is affected by a process that is not linked to translational selection. And indeed, there is strong evidence that isochores are

**\*For correspondence:**
Laurent.Duret@univ-lyon1.fr (LD);
marie.semon@ens-lyon.fr (MSé)

**Competing interests:** The authors declare that no competing interests exist.

the consequence of GC-biased gene conversion (gBGC), a form of segregation distortion that occurs during meiotic recombination and that favors the transmission of GC alleles over AT alleles (*Duret and Galtier, 2009*; *Munch et al., 2014*; *Williams et al., 2015*). This non-adaptive process leads to an increase in GC-content in regions of high recombination rate, which affects both coding and non-coding regions, including synonymous codon positions (*Galtier and Duret, 2007*; *Duret and Galtier, 2009*; *Glémin et al., 2015*).

In principle, this does not exclude that besides gBGC, codon usage bias might also be affected by translational selection. Interestingly, several studies have reported that human codon usage varies among genes expressed in different tissues or cell types (*Vinogradov, 2003*; *Plotkin et al., 2004*; *Gingold et al., 2014*). In particular, strong variations in SCU are observed among sets of human genes associated to different functional categories and notably between sets of genes involved in cellular proliferation or differentiation (*Gingold et al., 2014*). The relative abundance of tRNA varies also according to the proliferative or differentiation state of cells, which was logically interpreted in term of translational selection: different cell types express specific sets of genes whose coding sequence is co-adapted with specific pools of tRNAs (*Gingold et al., 2014*). If true, this has important implications regarding the role of translational regulation in determining cell fate (differentiation versus proliferation).

However, this interpretation stands in contradiction with two other studies examining tRNA abundance in mammals. First, although expression levels of individual tRNA genes vary substantially between tissue types and developmental stages in mice, the collective expression levels of isoacceptor tRNAs (which recognize the same codon) remain constant. Thus, the pool of available anticodons is stable throughout development (*Schmitt et al., 2014*). Second, in continuation to this work, a recent study specifically contrasted cells undergoing proliferation and those undergoing differentiation, and found no covariation of tRNA pool and codon usage between these cells (*Rudolph et al., 2016*). Both results are inconsistent with the differences in SCU between functional classes as being a consequence of translational selection.

The question of the relative contributions of adaptive and nonadaptive processes to variation in codon usage in mammals therefore remains open: on the one side, patterns of tRNA abundances do not fit with the translational selection model, but on the other side, the reason why codon usage varies among functional categories is not yet understood. Here, we examined the hypothesis that variation in codon usage might result from differences in transcription activity in meiotic cells. Indeed, it has been observed that intragenic recombination rate correlates negatively with expression level in the germline (*McVicker and Green, 2010*). It is therefore possible that differences in germline expression levels among functional categories induce differences in gBGC, and hence codon usage biases.

To test this hypothesis, we analyzed SCU among different functional categories of human genes, and investigated covariation with GC-content, recombination rate and expression patterns. We first show that the variation in codon usage among functional categories results from differences in GC content. Then, we propose a new test that demonstrates that variation in SCU is not associated with translational selection. Instead, SCU correlates with large-scale variation in genomic GC-content and with differences in intragenic recombination rate. In turn, the difference in intragenic recombination rate between functional categories is explained by their expression level in meiosis. Altogether, GC-content of non-coding regions and meiotic expression explain 70% of the variation in SCU of human genes. In the end, our results are fully consistent with the hypothesis that SCU is driven by gBGC, and not by translational selection. They indicate that the differences observed among functional categories reflect variation in long-term intragenic recombination rates, resulting from differences in meiotic expression levels.

## Results

### Variation in codon usage among functional categories results from differences in GC-content

To better understand the causes of the differences in codon usage between sets of genes involved in cellular proliferation and differentiation (reported by [*Gingold et al., 2014*]), we started by investigating the main factors that discriminate codon usage between functional categories in general. For

this purpose, we grouped genes per functional category (687 biological processes, associated to more than 40 genes in the Gene Ontology database), and computed codon frequencies for each of these gene sets. We used the classification proposed by *Gingold et al. (2014)* to distinguish GO gene sets associated to 'proliferation' or 'differentiation'. Variation in relative synonymous codon usage (RSCU; see Materials and methods) among GO gene sets was analyzed by Principal Component Analysis (PCA). The first principal component of this analysis segregates 'proliferation' (red dots) from 'differentiation' (blue dots) GO categories (*Figure 1A*). Thus, in agreement with *Gingold et al. (2014)*, synonymous codon usage clearly varies between functional categories in general, and between proliferation and differentiation in particular. Previous studies had shown that synonymous codon usage is correlated to GC content at third position of codons – termed GC3 (*Mouchiroud et al., 1988*). And indeed, we observed that the average GC3 of each GO gene set is perfectly correlated to their coordinates on the first PCA axis ($R^2 = 0.99$; *Figure 1B*). Hence, variation in SCU between functional categories is fully explained by variation in GC3.

On average, in our dataset, each gene is associated to nine GO biological processes. Many genes belong to more than one GO biological-process category, either because they have several functions (pleiotropy) or because these categories are nested from specific to broad functions. Hence, GO-terms are not independent. To avoid this redundancy, for the remainder of this study we switched from analyses at the level of GO gene sets to analyses at the level of individual genes (except when stated otherwise). Each gene was assigned with one of three categories based on their GO annotation: 1008 genes associated with 'proliferation', 2833 genes associated with 'differentiation', and 12,129 'other' genes unrelated to these key words (see Materials and methods). Genes associated to 'proliferation' are on average less GC-rich than genes associated to 'differentiation' (mean GC3 0.53 and 0.61 in the two subsets respectively). The two distributions of GC3 differ significantly from each other (t-test, p-value$<2.10^{-16}$), and their peaks coincide with each of the two modes observed for the rest of the genome (*Figure 1C*).

## Variation in synonymous codon usage is not driven by translational selection

We first investigated whether the observed variation in synonymous codon usage (i.e. variation in GC3) might be driven by translational selection. This model proposes that the relative usage of synonymous codons should co-vary with the abundance of their cognate tRNAs. A property of the tRNA gene repertoires allows us to test this hypothesis. The human genome contains 506 tRNA genes (decoding the 20 standard amino acids), corresponding to 48 different tRNA isoacceptors (*Chan and Lowe, 2016*). Among the 18 amino acids having two or more synonymous codons, 4 are decoded by a single tRNA isoacceptor (mono-isoacceptor amino acids: Phe, Asp, His and Cys), and the 14 other ones are decoded by several tRNA isoacceptors (multi-isoacceptors amino acids).

For multi-isoacceptors amino acids, the relative abundance of the different tRNA isoacceptors can vary among different cell types, and hence might covary with the relative synonymous codon usage of genes preferentially expressed in these cell types. For instance, let us consider Gln, which has two synonymous codons (CAG, CAA) that are decoded by two tRNA isoacceptors (respectively anticodons CTG and TTG). Let us consider a theoretical example of two cell types (say A and B) that differ in their relative tRNA abundance (CTG-tRNA being more abundant in A cells, and TTG-tRNA in B cells). According to the translational selection model, sets of genes that are over-expressed in A cells, should preferentially use the CAG codon, whereas genes that are over-expressed in B cells, should preferentially use the CAA codon. However, mono-isoacceptor amino acids are, by definition, decoded by a single tRNA isoacceptor and the relative tRNA abundance cannot vary across cell types. Hence, according to the translational selection model, the relative synonymous codon usage for mono-isoacceptor amino acids is not expected to vary among cell-specific gene sets. In other words, for mono-isoacceptor amino acids, variation in synonymous codon usage among GO gene sets cannot be explained by co-adaptation with the tRNA pool.

To test whether variation in synonymous codon usage was driven by translational selection, we computed synonymous codon usage (GC3) in GO gene sets, separately for codons corresponding to mono-isoacceptor amino acids and for codons corresponding to multi-isoacceptor amino acids. We observed that the range of variation in GC3 is very similar for mono- and multi-isoacceptor amino acids. Importantly, the two parameters are strongly correlated ($R^2 = 0.90$) (*Figure 1D*). This implies that GC3 variation is driven by a process that affects both mono-isoacceptor and multi-

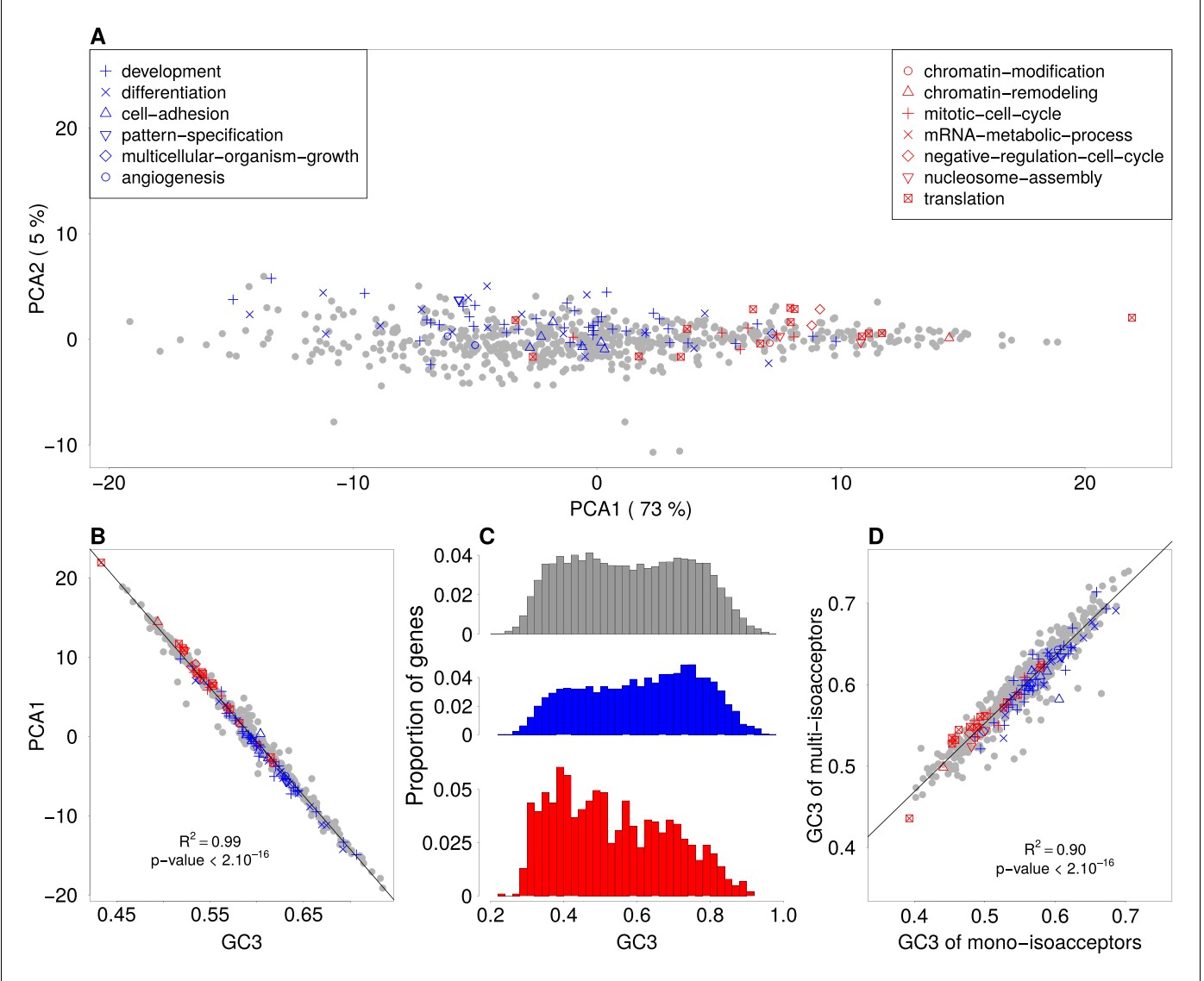

**Figure 1.** Variation in synonymous codon usage and in GC3 among functional categories. (**A**) Factorial map of the principal-component analysis of synonymous codon usage in GO functional categories in the human genome. Each dot corresponds to a GO gene set, for which the relative synonymous codon usage (RSCU) was computed. GO categories that are associated with 'differentiation' or with 'proliferation' are displayed in blue and in red, respectively. (**B**) Correlation between the RSCU of GO gene sets (first PCA axis) and their average GC-content at third codon position (GC3). (**C**) Distribution of GC3 of human protein coding genes. Red: 'proliferation' genes (N = 1,008); blue: 'differentiation' genes (N = 2,833); grey: other genes (N = 12,129). (**D**) Correlation between the GC3 of mono-isoacceptor amino acids and multi-isoacceptor amino acids. For each GO gene set, the average GC3 was computed separately for amino acids decoded by multiple tRNA isoacceptors (N = 14 multi-isoacceptor amino acids), and for those decoded by one single tRNA isoacceptor (mono-isoacceptor amino acids: Phe, Asp, His, Cys). Amino-acids encoded by a single codon (Met, Trp) were excluded.

DOI: https://doi.org/10.7554/eLife.27344.002

isoacceptor amino acids, and hence that this process is not related to variation in tRNA abundance. This observation holds true for all functional categories, including those associated to differentiation or proliferation (red and blue dots in *Figure 1D*).

## Impact of large-scale variation in genomic GC-content on synonymous codon usage

We observed that the GC3 of genes correlates with the GC-content of their flanking regions (*Figure 2A*, *Figure 2—figure supplement 1*, $R^2$ = 0.48, p-value<$2.10^{-16}$). This correlation is

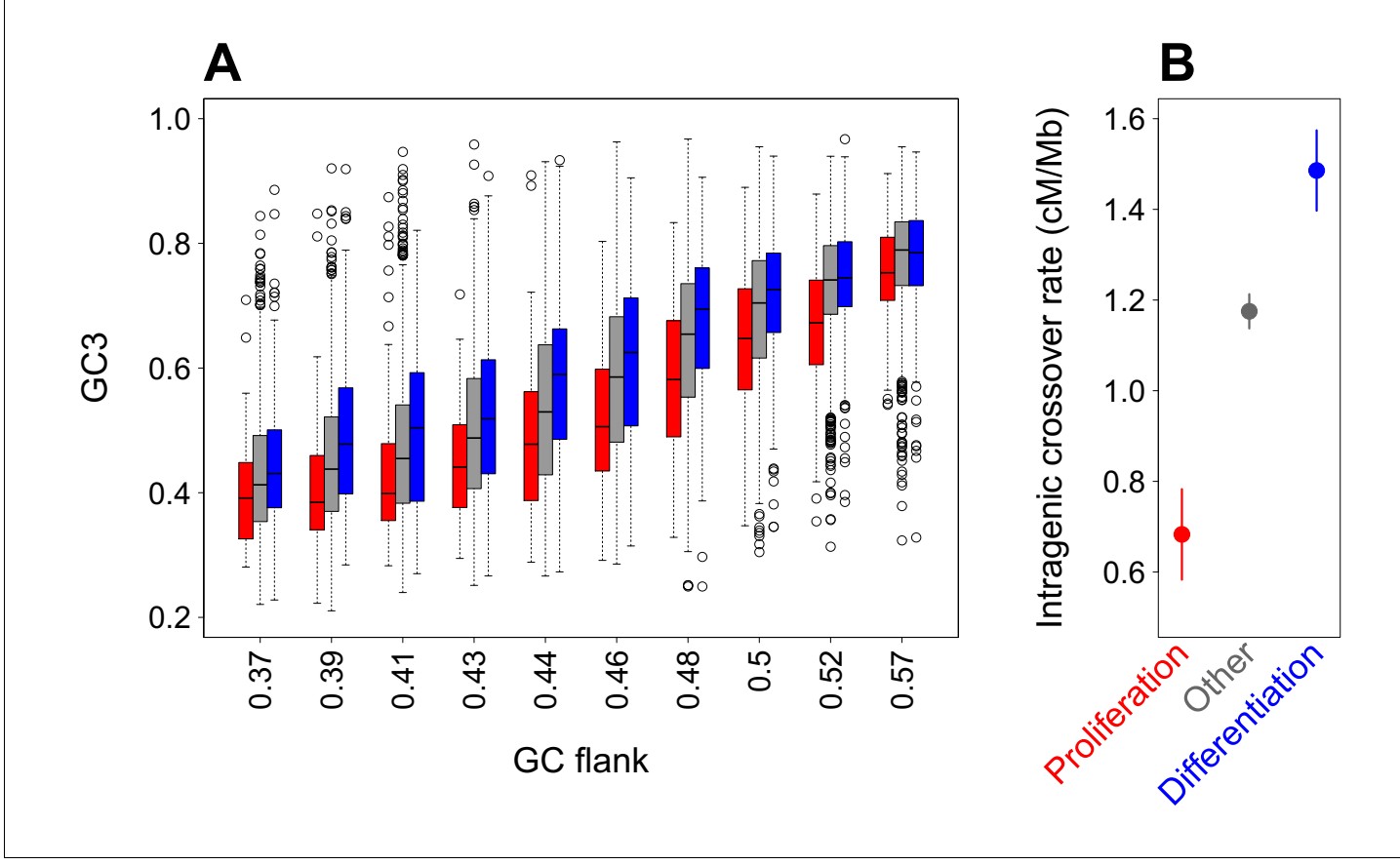

**Figure 2.** Difference in SCU between 'proliferation' and 'differentiation' genes is linked to variation in intragenic crossover rate, and not to their isochore context. (**A**) Variation in gene GC3 according to the GC content of their flanking region (GC-flank) in each functional category. Genes were first binned into 10 classes of equal sample size according to their GC-flank, and then split into three sets according to their functional category: 'proliferation' (red), 'differentiation' (blue), and 'other' genes (grey). Boxplots display the distribution of GC3 for each functional category within each GC-flank bin. (**B**) Mean sex-averaged intragenic crossover rate (HapMap) in each functional category. Error bars represent the 95% confidence interval of the mean.

DOI: https://doi.org/10.7554/eLife.27344.003

The following figure supplement is available for figure 2:

**Figure supplement 1.** Correlation between the GC3 of genes and the GC content of their flanking regions (GC-flank).

DOI: https://doi.org/10.7554/eLife.27344.004

observed for all genes, including the subsets of genes associated with 'proliferation' and 'differentiation' ($R^2$ = 0.48 and 0.46, all p-values<$2.10^{-16}$). Thus, variation in SCU between genes is to a large extent attributable to the GC-content of the genomic region in which they are located (the isochore effect). However, when the regional GC-content is controlled for, there remains a difference in GC3 between gene categories (*Figure 2A*): for a given regional GC-content, the GC3 of proliferation-associated genes is lower than that of differentiation or other genes. This difference is highly significant (*Figure 2A*, *Figure 2—figure supplement 1*, p-value<$2.10^{-16}$). This implies that the difference in synonymous codon usage between these gene categories does not result from a preferential location in different isochores.

## Variation in synonymous codon usage among functional categories correlates with differences in intragenic recombination rate

Previous studies have demonstrated that the evolution of GC-content along chromosomes is driven by meiotic recombination, both on a broad (Mb) scale (*Duret and Arndt, 2008*; *Munch et al., 2014*) and on a fine (kb) scale (*Clément and Arndt, 2013*; *Pratto et al., 2014*). There is now strong

evidence that this correlation between GC-content and recombination is caused by the process of GC-biased gene conversion (gBGC) which leads to increase the GC-content in regions of high recombination (*Galtier et al., 2001*; *Galtier and Duret, 2007*; *Duret and Galtier, 2009*; *Munch et al., 2014*; *Pratto et al., 2014*; *Williams et al., 2015*). Recombination rate varies along chromosomes, and notably tends to be lower within genes than in flanking regions (*Myers et al., 2005*; *McVicker and Green, 2010*). Interestingly, we observed that intragenic crossover rates (in cM/Mb) differ among the three sets of genes defined previously, and covary with their GC3: the average intragenic crossover rate is lower in 'proliferation' genes compared to other genes, whereas it is higher in 'differentiation' genes (*Figure 2B*; p-value of Kruskal-Wallis test <$2.10^{-16}$ as for all pair-wise Wilcoxon tests). These observations are therefore consistent with the hypothesis that differences in GC3 between 'differentiation' and 'proliferation' genes could also be driven by gBGC.

## The difference in intragenic recombination rate between functional categories is explained by their expression level in meiosis

*McVicker and Green (2010)* reported a negative correlation between intragenic recombination rate and meiotic gene expression level. We reevaluated this relationship using recently published high-resolution genetic maps (*Bhérer et al., 2017*), meiotic double-strand breaks (DSBs) maps (*Pratto et al., 2014*) and meiotic gene expression datasets (*Guo et al., 2015*; *Lesch et al., 2016*). These new data show that the relationship between crossover rate and meiotic gene expression is even stronger than initially reported: we observed that the crossover rate is 3.5 (males) to 5.4 (females) times lower in highly expressed genes (top 10%) compared to weakly expressed genes (bottom 10%) (*Figure 3A*, *Figure 3—figure supplement 3A,B*). This reduction in crossover rate is explained, at least in part, by a lower density of meiotic DSB hotspots within highly expressed genes (*Figure 3—figure supplement 3C*). In agreement with *Bhérer et al. (2017)*, we observed an elevation of crossover rate around transcription start sites, specifically in females (*Figure 4—figure supplement 1*). However, this peak is observed only in genes with low or medium meiotic expression level (*Figure 4*). Within genes with high meiotic expression level, we observed a strong reduction of crossover rate in both sexes, affecting the entire transcription unit, from the TSS to the polyadenylation site (*Figure 4*).

We also analyzed other RNA-seq data sets (either from single cells or bulk samples), covering a broad range of tissues/cell types: somatic or germ cells at different stages of developing male and female embryo (20 different conditions; [*Guo et al., 2015*]) and differentiated adult tissues (26 somatic tissues, plus testis, which contains a fraction of germ cells; [*Fagerberg et al., 2014*]). In agreement with *McVicker and Green (2010)*, we observed that the negative correlation between expression level and intragenic crossover rate is stronger in germ cells than in somatic samples (*Figure 3—figure supplement 1*), which indicates that recombination is associated with expression level, specifically in meiotic cells.

Many 'proliferation' genes are involved in basic cellular functions, and hence, tend to be expressed at relatively high levels in many tissues and at all developmental stages. In particular, most of these genes are highly expressed in meiotic cells: 65% of 'proliferation' genes are among the top 33% of genes with highest expression level (whereas only 11% are in the first tercile; *Figure 3—figure supplement 2*). Conversely, only 26% of 'differentiation' genes are highly expressed in meiotic cells, while 42% of are in the first tercile (*Figure 3—figure supplement 2*). This large proportion of 'proliferation' genes with high meiotic expression levels can therefore explain why they tend to have relatively low intragenic crossover rate (*Figure 2B*), and hence, given the gBGC process, why they tend to have a lower GC3 (*Figure 1C*). To further test whether these differences in expression patterns could account for the difference in GC3 between 'proliferation', 'differentiation' and 'other genes', we binned genes into three classes of increasing meiotic expression level. The distribution of GC3 is clearly shifted toward lower values for genes highly expressed at meiosis (top 33%), compared to genes weakly expressed (bottom 33%): the average GC3 is 0.51 in the 'high' category compared to 0.65 in the 'low' category (p-value<$2.10^{-16}$) (*Figure 3B*). However, there is no significant difference in the distribution of GC3 between 'proliferation' and 'differentiation' within bins of low or high expression (p-value=0.68 and 0.15 respectively). For the mid-expression bin, there is still a significant difference of GC3 between 'proliferation' and 'differentiation' (p-value=$3.2.10^{-8}$), potentially explained by differences in expression between categories within this bin.

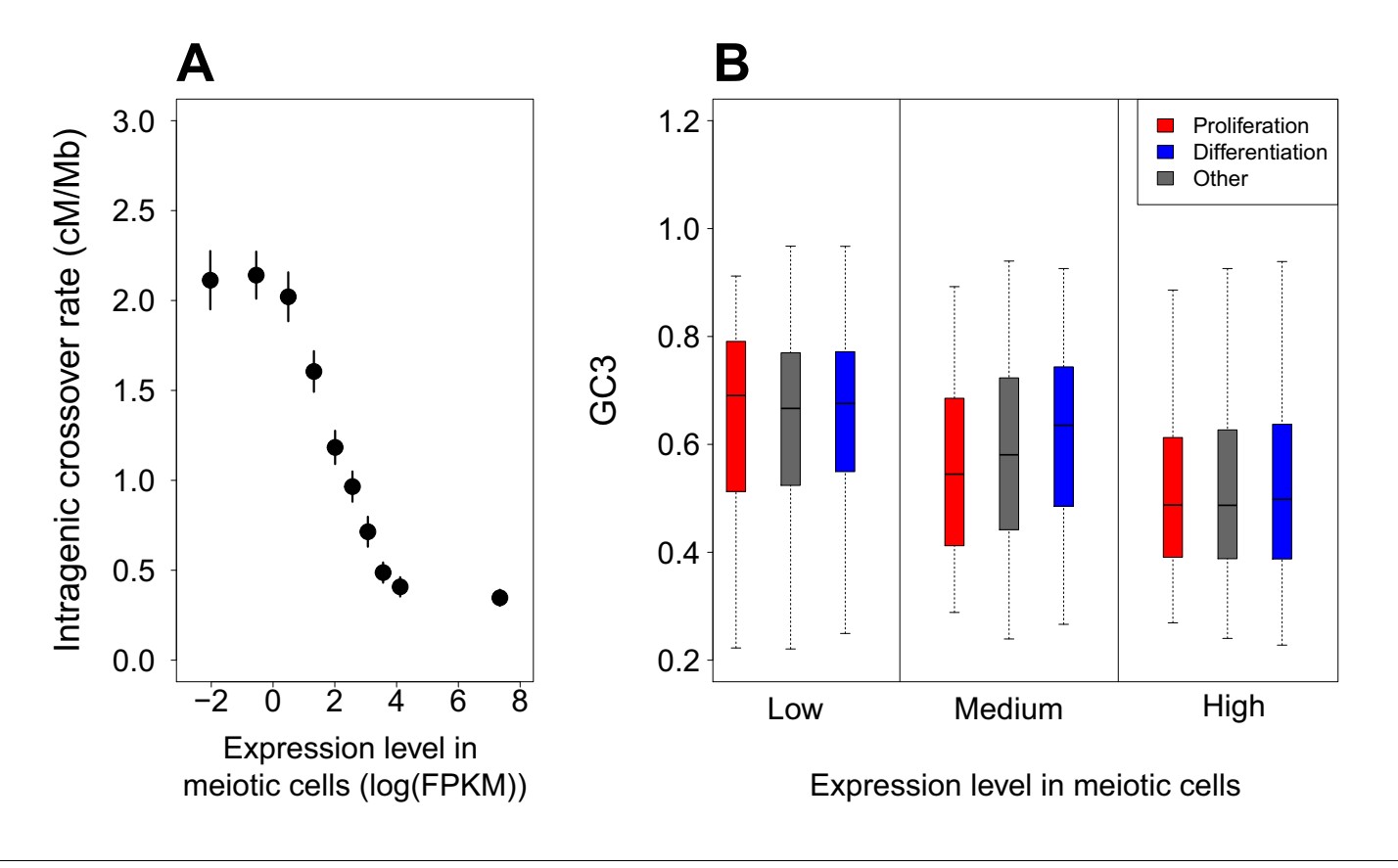

**Figure 3.** Variation in intragenic crossover rate and GC3 according to expression levels in meiotic cells. (**A**) Genes were classified according to their sex-averaged expression level in meiotic cells into 10 bins of equal sample size. The mean sex-averaged intragenic crossover rate (HapMap) was computed for each bin. Error bars represent the 95% confidence interval of the mean. Similar results were obtained when analyzing sex-specific crossover rates and expression levels or when using DSB maps to measure of recombination rate (*Figure 3—figure supplement 3*). (**B**) Variation in GC3 according to meiotic expression levels. Genes were first binned into 3 classes of equal sample size according to their sex-averaged expression level in meiotic cells (low:<3.07 FPKM; high:>22.68 FPKM: medium: the others), and then split into three sets according to their functional category: 'proliferation' (red), 'differentiation' (blue), and 'other' genes (grey). Boxplots display the distribution of GC3 for each functional category within each expression bin.
DOI: https://doi.org/10.7554/eLife.27344.005

The following figure supplements are available for figure 3:

**Figure supplement 1.** Differential intragenic crossover rate between lowly and highly expressed genes in adult tissues and in individual embryonic cells.
DOI: https://doi.org/10.7554/eLife.27344.006

**Figure supplement 2.** Comparison of the distribution of meiotic gene expression levels for 'proliferation', 'differentiation' and other genes.
DOI: https://doi.org/10.7554/eLife.27344.007

**Figure supplement 3.** Variation in intragenic recombination rate and GC3 according to expression levels in meiotic cells.
DOI: https://doi.org/10.7554/eLife.27344.008

Thus, most of the difference in synonymous codon usage between functional categories (*Figure 1C*) disappears once level of expression during meiosis is controlled for (*Figure 3B*).

Thus, differences in synonymous codon usage among gene categories in human can be explained through the following causative chain: (i) The set of 'proliferation' genes is enriched in genes highly expressed in meiosis. (ii) Because high expression at meiosis is associated with a decreased rate of recombination, intragenic recombination rates are lower in the 'proliferation' set. (iii) In turn, reduced intragenic recombination diminishes the effect of gBGC on exon base composition, and hence GC3 is lower in the set 'proliferation' compared to 'differentiation'.

To check whether this cascade of effects fully recapitulates the difference in synonymous codon usage between 'proliferation' and 'differentiation', we investigated whether differences in SCU

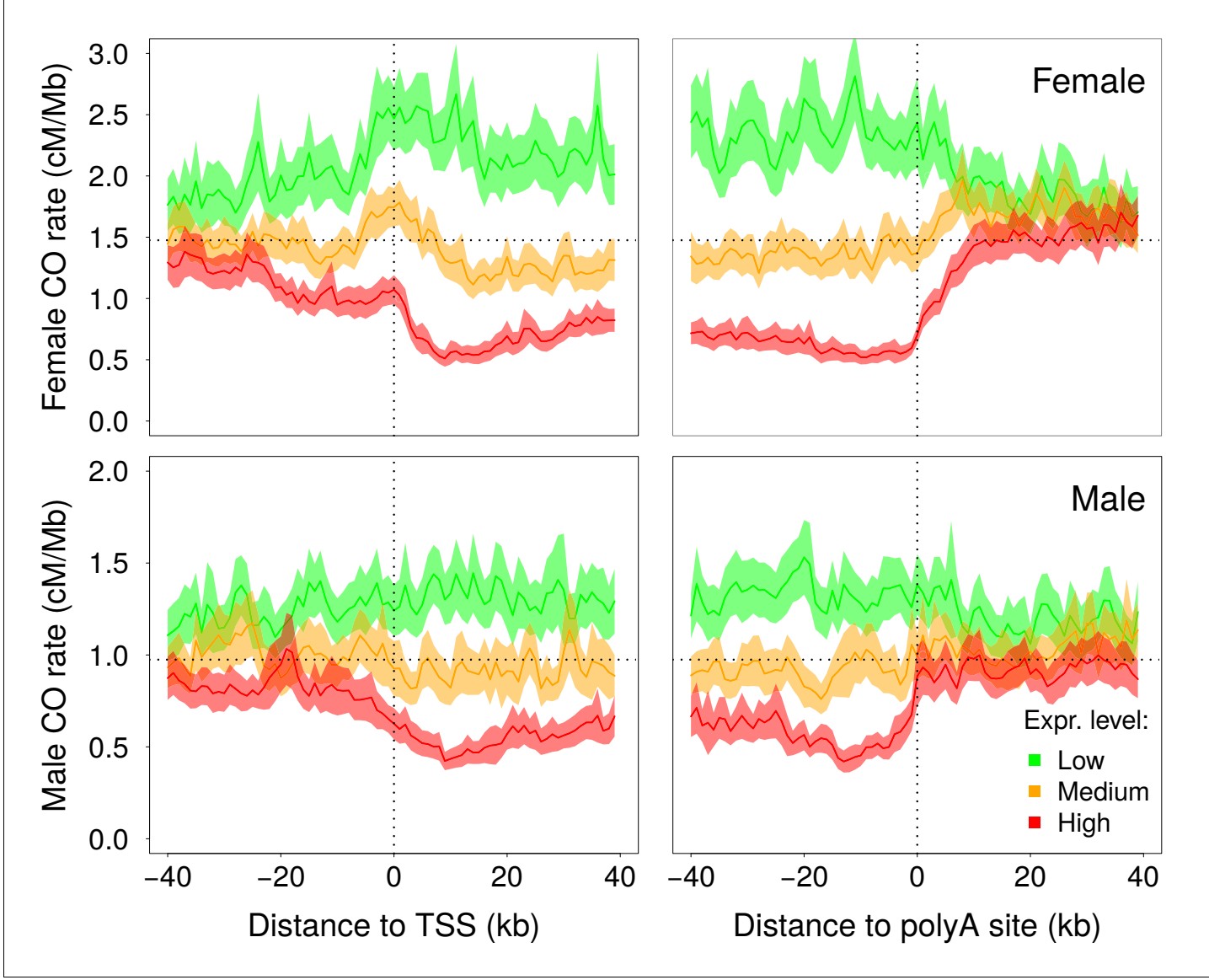

**Figure 4.** Variation in crossover rate as a function of the distance to transcription start site (TSS) and to the polyadenylation site, and according to meiotic expression level. Autosomal genes longer than 5 kb (N = 15,055) were classified into three bins of equal sample size according to their expression level in female (top panels) or male meiosis (bottom panels): low (green), medium (orange) and high (red) expression level. Sex-specific crossover rates were measured in 1 kb-long non-overlapping windows. Shaded area represent the 95% confidence interval of the mean.

DOI: https://doi.org/10.7554/eLife.27344.009

The following figure supplements are available for figure 4:

**Figure supplement 1.** Variation in crossover rate as a function of the distance to transcription start site (TSS) and to the polyadenylation site.
DOI: https://doi.org/10.7554/eLife.27344.010

**Figure supplement 2.** Variation in DSB hotspot density as a function of the distance to transcription start site (TSS) and to the polyadenylation site, and according to meiotic expression level.
DOI: https://doi.org/10.7554/eLife.27344.011

between functional categories are driven by expression level in cells undergoing meiosis, rather than by expression level in another cell type or tissue. We examined the relationship between GC3 and expression levels in a broad panel of cell and tissue conditions (*Figure 5*). As predicted by our model, expression levels in germ cells, either from single-cell samples or from testis (which contains germ cells) are better predictors of GC3 than expression in all other somatic tissues. Strikingly, the levels of expression in primary germ cells is, on average, twice as informative than expression in

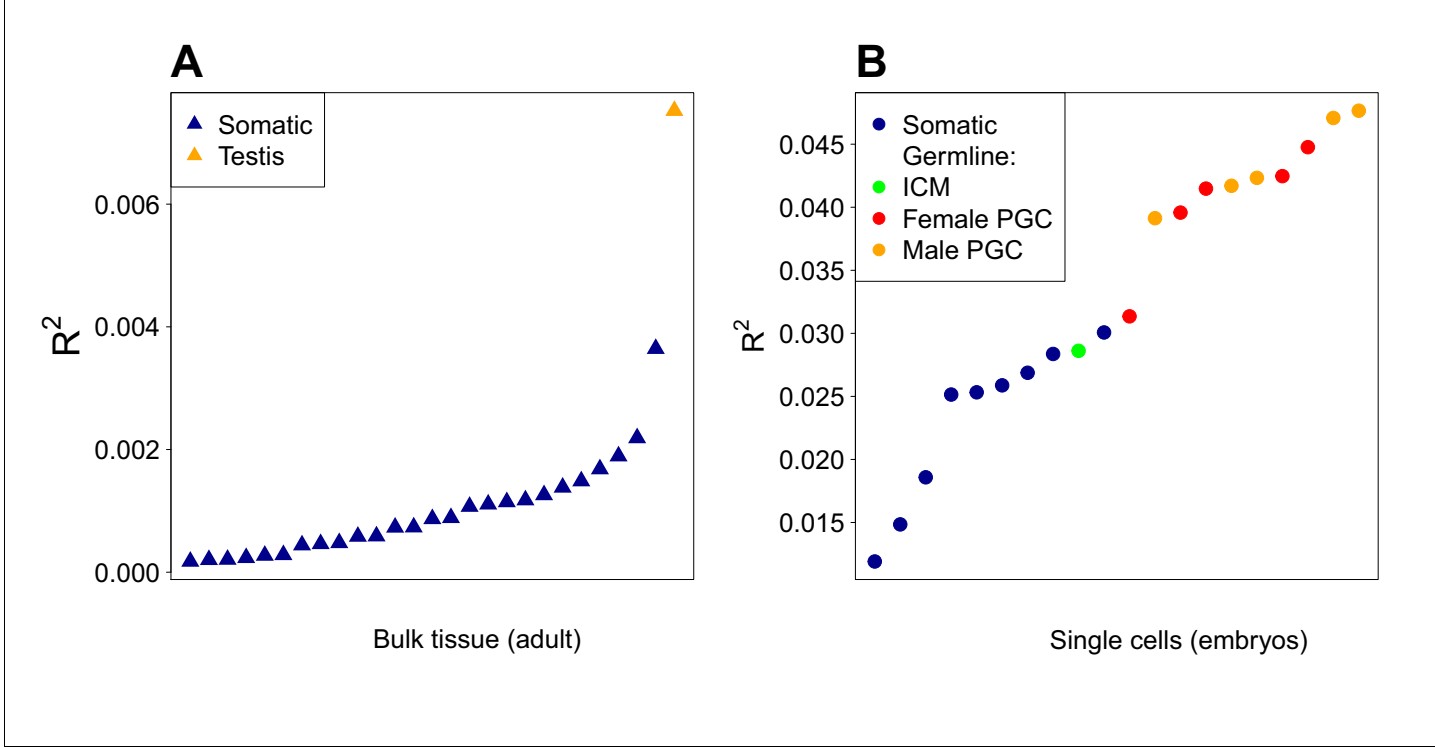

**Figure 5.** Correlation between expression level and GC3 in a panel of tissues and cell types. (A) Bulk adult tissues data (*Fagerberg et al., 2014*) and (B) early embryo single-cell data (*Guo et al., 2015*). These two subsets were obtained via very different protocols, which prevents direct cross-comparisons. Samples are sorted by increasing correlation coefficient ($R^2$) between expression levels and GC3 (NB: all correlations are negative). Samples containing somatic cells are shown in blue; male germ cells in orange (testis or single cell) and female germ cells in red (PGC: primordial germ cells). The green point corresponds to cells from the inner cell mass (ICM) of the blastocysts, i.e. pluripotent cells from an early stage of development preceding the differentiation of germ cells.

DOI: https://doi.org/10.7554/eLife.27344.012

somatic cells taken at comparable stage of development (*Figure 5B*). Among all individual samples, the strongest correlation between GC3 and expression level was found in male meiotic cells (pachytene spermatocytes, $R^2 = 6.3\%$, p-value$<2.10^{-16}$). Female meiotic cells (primordial germ cells, PGC 17 W) showed a similar correlation level ($R^2 = 4.0\%$, p-value$<2.10^{-16}$). As expected, the correlation is even stronger with sex-averaged meiotic expression level ($R^2 = 8.6\%$, p-value$<2.10^{-16}$). Hence, these results confirm that the cell type for which gene expression level is the best predictor of GC3 (and therefore SCU) corresponds to meiotic cells.

## GC-content of non-coding regions and meiotic expression explain 70% of the variation in synonymous codon usage of human genes

Meiotic expression is associated with a deficit of recombination rates all along the gene (*Figure 4*). Thus, the expression pattern is expected to affect gBGC intensity (and hence the GC-content) both in exons and in introns. Consistent with that prediction, the GC3 of human genes is strongly correlated to the GC-content of their introns (GCi, $R^2 = 62.7\%$, p-value$<2.10^{-16}$). We build a linear model to quantify the relative contribution of the different parameters that covary with the GC3 of human genes (GCi, GC-flank, intragenic crossover rate, meiotic expression level, and 'proliferation' or 'differentiation' functional category). The analysis of variance demonstrates that GCi is by far the best predictor of GC3, but GC-flank, intragenic crossover rate and gene expression level during meiosis, also significantly improve the model (by 1%, 4% and 1.4%, respectively, *Table 1*, ANOVA, p-values$<2.10^{-16}$). The integration of a categorical variable 'differentiation' versus 'proliferation' in the model significantly improves the model but its quantitative influence is minor (0.1%, p-value$<2.10^{-16}$, *Table 1*). Altogether, 68.2% of the variance in GC3 among human genes can be explained by the first four parameters (GCi, GC-flank, intragenic crossover rate, meiotic expression).

**Table 1.** Analysis of the variance of GC3 among individual genes.

Variables included in the linear model are: GC-content of introns (GCi), GC-content of flanking regions (GC-flank), HapMap sex-averaged intragenic crossover rate (log scale), sex-averaged meiotic gene expression level (log scale) and functional category ('differentiation', 'proliferation' and 'other'). Pairwise correlations (pairwise $R^2$) were computed between GC3 and each of the other variables. Correlations of the model (model $R^2$) were computed by adding variables sequentially.

| GC3 predictors | Pairwise $R^2$ | p-value | Model $R^2$ | F statistic | p-value |
|---|---|---|---|---|---|
| GCi | 62.7% | $<2.10^{-16}$ | 62.7% | 30232.4 | $<2.10^{-16}$ |
| GC-flank | 48.1% | $<2.10^{-16}$ | 62.9% | 126.8 | $<2.10^{-16}$ |
| Intragenic crossover rate | 12.8% | $<2.10^{-16}$ | 66.8% | 1453.3 | $<2.10^{-16}$ |
| Expression level in meiosis | 8.3% | $<2.10^{-16}$ | 68.2% | 875.7 | $<2.10^{-16}$ |
| Functional category | 1% | $<2.10^{-16}$ | 68.3% | 30.43 | $<2.10^{-16}$ |

DOI: https://doi.org/10.7554/eLife.27344.013

Adding interaction terms to the linear model gives very similar results (70.4% variance explained, same levels of significance for all variables).

## Discussion

### Biased gene conversion drives codon usage in humans

In the human genome, gene sets that belong to different functional categories differ by their synonymous codon usage. Initially this pattern has been interpreted as evidence that the translation program was under tight control, notably to ensure a precise regulation of genes involved in cellular differentiation or proliferation (*Gingold et al., 2014*). According to this model, selection should optimize the match between the SCU of genes and tRNA abundances in the cells where they are expressed. However, the comparison of synonymous codon usage for amino acids with single or multiple tRNA isoacceptors (*Figure 1D*) shows that the difference in SCU between functional categories does not result from constraints linked to tRNA abundance. In fact, variation in synonymous codon usage among functional categories is explained by one single dominant factor: the GC-content at third codon position (*Figure 1B*). The GC3 of human genes is strongly correlated to the GC-content of their introns and flanking regions (*Table 1*). This implies that variation in SCU results from a process that affects both coding and non-coding regions (including non-transcribed intergenic regions), and hence that it is not related to the process of translation. In fact, this observation invalidates all the models that assume that SCU is driven by a selective pressure acting on RNAs (not only translational selection, but also selection on mRNA processing, structure or stability).

Many lines of evidence indicate that large-scale variation in GC-content along chromosomes (isochores) is driven by the gBGC process, both in mammals and birds. First, there is direct evidence that recombination favors the transmission of GC-alleles over AT-alleles during meiosis (*Odenthal-Hesse et al., 2014*; *Arbeithuber et al., 2015*; *de Boer et al., 2015*; *Williams et al., 2015*; *Smeds et al., 2016*). Second, the analysis of polymorphism and divergence at different physical scales (from kb to Mb) showed that recombination induces a fixation bias in favor of GC alleles (*Duret and Arndt, 2008*; *Clément and Arndt, 2013*; *Munch et al., 2014*; *Pratto et al., 2014*; *Weber et al., 2014*; *Glémin et al., 2015*; *Singhal et al., 2015*). Third, the gBGC model predicts that the GC-content of a given genomic segment should reflect its average long-term recombination rate over tens of million years (*Duret and Arndt, 2008*). Consistent with this prediction, analyses of ancestral genetic maps in the primate lineage revealed a very strong correlation between long-term recombination rates (in 1 Mb long windows) and stationary GC-content – $R^2 = 0.64$; (*Munch et al., 2014*). The strong correlation between GC3 and GC-flank therefore implies that variation in synonymous codon usage is primarily driven by large-scale variation in long-term recombination rate.

Besides these regional fluctuations, recombination rates also vary at finer scale. In particular, recombination rates tend to be reduced within human genes compared to their flanking regions (*Myers et al., 2005*), and this decrease depends on the level of expression of genes during meiosis (*McVicker and Green, 2010*) – see also *Figure 3A* and *Figure 4*. Hence, the gBGC model predicts

that the GC3 of a gene should depend not only of the long-term recombination rate of the region where it is located, but also on its specific pattern of expression. And indeed, we observed that the difference in synonymous codon usage between 'proliferation' and 'differentiation' genes is not due to their preferential location in different classes of isochores, but to the fact that 'proliferation' genes tend to be expressed a high level in meiotic cells, and therefore to have a reduced intragenic recombination rate (*Figures 2* and *3*).

To test whether this observation holds true for other functional categories, we measured the average GC3, intragenic crossover rate and meiotic expression level of each GO gene set. As predicted by the gBGC model, we observed a strong correlation between GC3 and the average intragenic

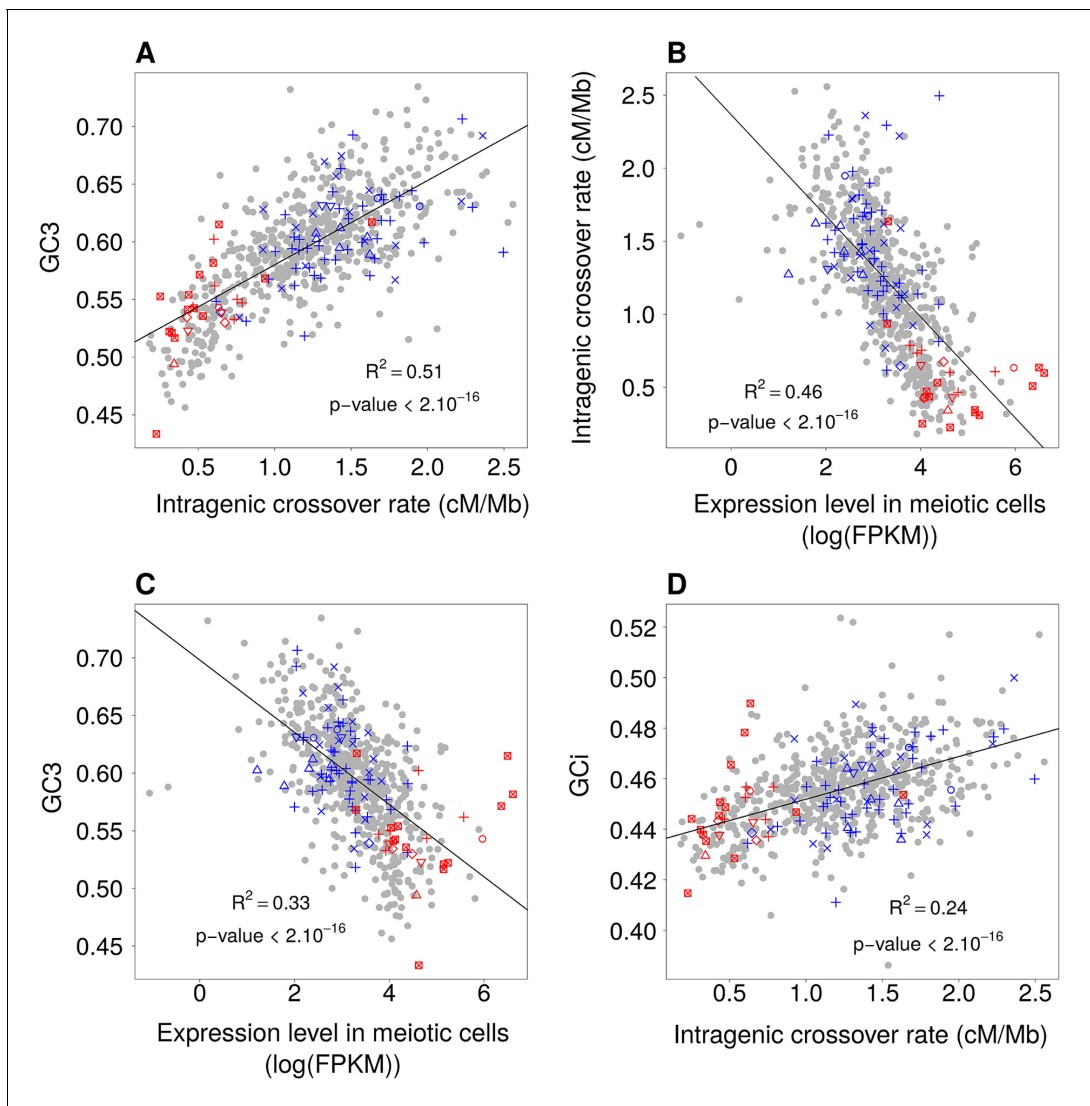

**Figure 6.** Relationships between GC-content, intragenic crossover rates and meiotic expression levels (sex-averaged) among functional gene categories. Average values of these parameters were computed for each GO gene set. We then measured correlations between these parameters: (**A**) Mean GC3 vs. mean sex-averaged intragenic crossover rate (HapMap). (**B**) Mean intragenic crossover rate vs. mean expression level in meiotic cells. (**C**) Mean GC3 vs. mean expression level in meiotic cells. (**D**) Mean intronic GC-content (GCi) vs. mean intragenic crossover rate. GO gene sets associated to 'proliferation' (red) or 'differentiation' (blue) are displayed as in *Figure 1*. Similar results were obtained when analyzing separately expression levels in female or male meiosis (*Figure 6—figure supplement 1*).

DOI: https://doi.org/10.7554/eLife.27344.014

The following figure supplement is available for figure 6:

**Figure supplement 1.** Relationships between expression levels in female or male meiotic cells and GC3 and intragenic crossover rates.

DOI: https://doi.org/10.7554/eLife.27344.015

crossover rate of GO gene sets ($R^2$ = 0.51, *Figure 6A*). The variance in intragenic crossover rate, in turn, is very well explained by differences in meiotic expression levels among functional classes ($R^2$ = 0.46, *Figure 6B*). As mentioned previously, these correlations measured on gene concatenates should be interpreted with caution because the different points are not independent (a same gene can belong to different GO categories). However, this analysis clearly shows that a large fraction of the variance in SCU observed among GO gene sets can be explained by variation in gBGC intensity, caused by variation in intragenic crossover rates, linked to differences expression patterns (*Figure 6C*). In agreement with the gBGC model, the intragenic crossover rate correlates with the base composition of the entire gene, including introns (*Figure 6D*). This observation clearly invalidates the hypothesis that the observed differences in SCU among functional categories might be driven by selection on codon usage.

In summary, the SCU of individual genes depends primarily on the isochore in which they are located (i.e. large-scale long-term variation in recombination rate), and secondarily on their meiotic expression level (which affects locally the intragenic recombination rate) (*Table 1*). In gene set analyses, the variance in SCU explained by expression (*Figure 6*) appears much stronger than in individual genes analyses (*Table 1*). This is due to the fact that in gene set analyses, SCU is averaged over a large number of genes, located in different isochores, which leads to decrease the isochore effect among functional categories (and hence mechanically increase the fraction of the variance explained by expression). Overall, the different variables linked to the intensity of gBGC explain 70% of the variance in GC3 of individual genes (*Table 1*). In other words, the gBGC model can account for most of the variation in synonymous codon usage in the human genome.

It should be noted that co-variation between SCU and expression is generally considered as a typical signature of translational selection and is often used to predict optimal codons (*Duret, 2002*; *Plotkin et al., 2004*; *dos Reis and Wernisch, 2009*). However, as shown here, such correlations can also emerge as a result of a non-adaptive process. Given that gBGC is widespread in eukaryotes (*Mancera et al., 2008*; *Capra and Pollard, 2011*; *Pessia et al., 2012*; *de Boer et al., 2015*; *Williams et al., 2015*; *Smeds et al., 2016*), it appears essential to take this process into account to interpret variation in synonymous codon usage (and more generally in base composition) among genes.

## Relationship between meiotic expression and recombination

The reason why intragenic recombination rate correlates negatively with meiotic expression level is not known. In human and mice, the location of recombination hotspots is determined by PRDM9, a Zn-Finger DNA-binding protein with histone H3 lysine four trimethylation (H3K4me3) activity. PRDM9 is expressed during early meiosis and marks sites where DSBs are afterwards introduced by Spo11 (for review, see *Baudat et al., 2013*). These DSBs are then repaired by homologous recombination, forming either crossovers, the reciprocal exchanges of genetic material between parental chromosomes, or noncrossovers. Knockout experiments in mice have demonstrated that PRDM9 targets recombination away from active promoters (*Brick et al., 2012*). The analyses of male DSB maps suggests that PRDM9 plays the same role in humans: we observed a deficit of DSB hotpots around the transcription start site (TSS), specifically within genes that are highly expressed in meiotic cells (*Figure 4—figure supplement 2*). The decrease in recombination rate within highly expressed genes is however not restricted to the promoter region: in both sexes, there is a strong deficit of crossovers within the entire transcription unit, from the TSS to the polyadenylation site (*Figure 4*). In species that lack Prdm9 (such as dogs, birds, arabidopsis or yeast), recombination hotspots are strongly enriched in active promoters (*Auton et al., 2013*; *Choi et al., 2013*; *Singhal et al., 2015*; *Lam and Keeney, 2015*), which indicates that there is no mechanistic incompatibility between recombination and transcription activity in meiotic cells. However there is evidence that in highly expressed genes, H3K36me3 marks trigger DNA methylation in the gene body, and thereby prevent spurious transcription initiation (*Neri et al., 2017*). It is therefore possible that the peculiar chromatin state of highly expressed genes also interferes with the binding of PRDM9 (or with its histone modification activity), and thereby decrease the rate of DSB formation within the transcription unit. Consistent with this hypothesis, we observed a deficit in male DSB hotspot density along the transcription unit of highly expressed genes (*Figure 4—figure supplement 2*). This difference in DSB rates is, however, much less pronounced than the difference in male crossover rates (*Figure 4*; *Figure 3—figure supplement 3*). Furthermore, the profile of DSB hotspot density in highly expressed genes differs

from that of crossover rates, with a strong deficit around the TSS and an excess around the polyade-nylation site (*Figure 4—figure supplement 2*), whereas the deficit in male crossovers is more uni-form along the transcription unit (*Figure 4*). This suggests that the differences in crossover profiles observed between highly and weakly expressed genes might also reflect differences in the way recombination events are resolved (crossover vs. non-crossovers).

## gBGC precludes selection on translation efficiency in humans

There is a clear evidence that the usage of synonymous codons is under selective pressure in some metazoan species (such as drosophila or nematode), which implies that it has a significant impact on the fitness of organisms – for review, see (*Duret, 2002*; *Chamary et al., 2006*; *Plotkin and Kudla, 2011*). It is *a priori* expected that codon usage should also affect translation efficiency (speed and accuracy) in mammals. However, our results show that selection on codon usage is not strong enough to counteract the impact of gBGC. In principle, this does not exclude the hypothesis that the human genome might be subject to selection for translational efficiency: even if the GC-content of genes is driven by non-adaptive processes, there might be a selective pressure on the expression of tRNA genes to match the demand in synonymous codon usage. However, recent analyses of tRNA isoacceptors pools found no evidence for such variation (*Schmitt et al., 2014*; *Rudolph et al., 2016*). Moreover, we argue here that the peculiar base composition landscape induced by gBGC in the genomes of mammals and birds makes it impossible to match the tRNA pool to the demand in codon usage. Indeed, large-scale variation in recombination rates along the genome causes very strong variation in GC3 among genes, and this, regardless of their functional category. In particular, 'proliferative' genes, which are involved in basic cellular process, and are expressed at high levels in most tissues, show a very strong heterogeneity in GC3 (from 20% to almost 100%; *Figure 1C*). This implies that in any given cell, the set of highly expressed genes will show a very heterogeneous usage of synonymous codons. Hence, whatever the pool of tRNA available in that cell, there will be a large fraction of genes with a codon usage that does not match tRNA abundance. In other words, the heterogeneity of synonymous codon usage in mammalian genomes reflects a non-optimal situa-tion, caused the gBGC process, in which it is not possible to adapt the tRNA pool to the demand in codon usage of the transcriptome of any cell type.

## Materials and methods

### Human protein coding genes

For each of the human protein coding genes in the Ensembl (RRID: SCR_002344) release 83 (*Yates et al., 2016*); assembly GRCh38.p5), we identified a canonical transcript as defined in http://www.ensembl.org/Help/Glossary?id=346 (PERL script available in supplementary material). Mito-chondrial genes were excluded from this analysis. Sequences of the remaining 19,766 canonical tran-scripts together with exons coordinates, were downloaded through the BioMart query interface (*Smedley et al., 2015*)(RRID: SCR_010714).

### Recombination rates

Sex-specific crossover rates were measured using pedigree-based genetic maps (*Bhérer et al., 2017*). For sex-averaged crossover rates, we used the HapMap genetic map (*Frazer et al., 2007*) (RRID: SCR_002846), which is based on the analysis of linkage disequilibrium in human populations, and provides a higher resolution than pedigree-based genetic maps.

The density in DSB hotspots along genes was measured using the map of DSB hotspots (targeted by Prdm9 alleles A, B or C) identified by DMC1-ChipSeq experiments in male meiotic cells (*Pratto et al., 2014*).

### Definition of functional categories

The GO Term Accessions and GO domain were retrieved from Ensembl version 83 for the 19,766 genes. We retrieved biological process GO terms, counted the number of genes associated to each GO term and kept the ones that include at least 40 genes, except GO:0005515 that is too general to be informative ('protein binding' GO set, which includes 14,542 genes). This led to a final list of 687 GO gene sets. For each gene set, we concatenated coding sequences to compute the total

codon usage, the relative synonymous codon usage (RSCU) and GC-content, and we also computed the average intragenic crossover rate and average expression levels (see below). The RSCU of a given codon corresponds to its frequency, normalized by its expected frequency if all corresponding synonymous codons were equally used (*Sharp et al., 1986*). For a given amino acid (*x*), encoded by $n_x$ synonymous codons, the RSCU of its codon *y* is given by:

$$RSCU_{xy} = C_{xy}/(A_x/n_x)$$

where $C_{xy}$ is the number *y* for amino acid *x*, $A_x$ is the total number of occurrence of codons for the amino acid *x*.

Following the classification used by *Gingold et al. (2014)*, we further defined two broad functional categories: 'proliferation' and 'differentiation'. GO terms containing the following keywords were associated to 'proliferation': 'Chromatin modification', 'chromatin remodeling', 'mitotic cell cycle', 'mRNA metabolic process', 'negative regulation of cell cycle', 'nucleosome assembly', 'translation'. GO terms containing the following keywords were associated to 'differentiation': 'Development', 'differentiation', 'cell adhesion', 'pattern specification', 'multicellular organism growth', 'angiogenesis'. Please note that GO terms corresponding to negative effects were excluded where appropriate (e.g. 'negative regulation of proliferation' was not included in the 'proliferation' category). Complete lists of GO terms are available in the supplementary material.

## Analyses of individual genes

We also measured the codon usage of individual genes, to analyze covariations with their GC-content, expression levels and sex-averaged intragenic crossover rate (HapMap). Owing to the low SNP density in human populations, the resolution of recombination maps is limited to about 5 kb (*Myers et al., 2005*). Because we investigate the relationship between GC3 and intragenic crossover rate, we selected genes that are long enough to measure recombination, that is at least 5 kb long (N = 16,223 genes).

We defined three non-overlapping classes of genes according to their GO category: genes associated to at least one of the 'proliferation' GO terms (N = 1,008), genes associated to 'differentiation' GO terms (N = 2,833) and other genes (N = 12,129). A group of 253 genes that were associated to both 'proliferation' and 'differentiation' GO terms were discarded from further analyses. The final dataset used in our analyses included 15,970 genes. In this dataset, there were 15,848 genes that contain at least one intron and for which we computed the GC content of intronic regions. The analyses of sex-specific crossover rates and of DSB hotspot densities (*Figure 4*; *Figure 4—figure supplement 2*) were based on 15,055 autosomal genes.

## Expression data

Gene expression levels were collected from three publicly available human RNA-seq experiment datasets. The first one includes 27 differentiated adult tissues (*Fagerberg et al., 2014*; *Kryuchkova-Mostacci and Robinson-Rechavi, 2015*); EBI accession number E-MTAB-1733. We downloaded normalized expression levels, already averaged across replicates, from (*Fagerberg et al., 2014*; *Kryuchkova-Mostacci and Robinson-Rechavi, 2015*) (see supplementary information). The second one is based on single-cell RNA-seq analysis, and includes 20 samples, corresponding to inner cell mass (ICM) of the blastocysts, and to primordial germ cells (PGC) and somatic cells, from male and female embryos at different development stages (4, 7 or 8, 10, 11 and 17 or 19 weeks, (*Guo et al., 2015*) GEO accession number GSE63818). We downloaded normalized expression levels from their dataset of pool-split PGCs (for more details see supplementary information). Female 17 weeks PGCs are entered in meiosis (*Guo et al., 2015*). This sample was therefore taken as representative of the transcriptome of meiotic cells in female. The third dataset corresponds to human male germ cells at pachytene spermatocytes (i.e. cells entering meiosis) and at round spermatids stages (post meiotic stage) ([*Lesch et al., 2016*]; GEO accession number GSE68507, human RNA expression datasets GSM1673959, GSM1673963, GSM1673967, GSM1673971, GSM1673975 and GSM1673978). Guo and Lesch datasets include several replicates for each sample. We therefore computed the average expression levels over all replicates for each sample. The sex-averaged meiotic expression level was estimated by computing the mean of expression levels in female 17 weeks PGCs (*Guo et al., 2015*) and male spermatocytes or spermatids (*Lesch et al., 2016*). The correspondence between gene

expression datasets and codon usage tables was based on Ensembl gene identifiers (Fagerberg and Lesch datasets), or on gene names (Guo dataset). In total, our analyses of expression levels were based on 15,305 genes (665 genes were absent from the Guo dataset).

## Statistical analysis

Unless stated otherwise, reported $R^2$ values correspond to Pearson correlation tests. R version 3.2.2 (*Core Team R, 2015*) was used with Base package for statistical tests and graphics, plus ade4 library (*Dray and Dufour, 2007*) for PCA analysis. The data and R scripts, which permit to reproduce the figures and tests presented here, are provided in the supplementary material.

## Supplementary information

Supplementary materials with R scripts and supplementary methods are available at: http://doi.org/10.5281/zenodo.835063 (*Pouyet et al., 2017*).

## Acknowledgement

This work was supported by French National Research Agency (ANR) grant DaSiRe (ANR-15-CE12-0010-01/DaSiRe) and the "appel d'offre fond recherche-projets emergents" of the ENS de Lyon. This work was performed using the computing facilities of the CC LBBE/PRABI. FP received a doctoral scholarship from Ecole Normale Supérieure de Lyon (http://www.ens-lyon.eu/). We thank Gaël Yvert for initiating the discussion and Adam Eyre-Walker and Vincent Daubin for helpful suggestions on a first version of our manuscript.

## Additional information

### Funding

| Funder | Grant reference number | Author |
|---|---|---|
| Agence Nationale de la Recherche | ANR-530 15-CE12-0010-01/DaSiRe | Laurent Duret |
| École Normale Supérieure de Lyon | Projet Emergent | Marie Sémon |

The funders had no role in study design, data collection and interpretation, or the decision to submit the work for publication.

### Author contributions

Fanny Pouyet, Conceptualization, Data curation, Formal analysis, Methodology, Writing—original draft; Dominique Mouchiroud, Conceptualization, Supervision, Methodology; Laurent Duret, Conceptualization, Data curation, Formal analysis, Supervision, Methodology, Writing—original draft; Marie Sémon, Conceptualization, Formal analysis, Supervision, Methodology, Writing—original draft

### Author ORCIDs

Fanny Pouyet http://orcid.org/0000-0001-5614-6998
Laurent Duret http://orcid.org/0000-0003-2836-3463
Marie Sémon http://orcid.org/0000-0003-3479-7524

### Decision letter and Author response

Decision letter https://doi.org/10.7554/eLife.27344.027
Author response https://doi.org/10.7554/eLife.27344.028

## Additional files

### Supplementary files

- Transparent reporting form

DOI: https://doi.org/10.7554/eLife.27344.016

## Major datasets

The following previously published datasets were used:

| Author(s) | Year | Dataset title | Dataset URL | Database, license, and accessibility information |
|---|---|---|---|---|
| Guo F, Yan L, Guo H, Li L, Hu B, Zhao Y, Yong J, Hu Y, Wang X, Wei Y, et al | 2015 | The transcriptome and DNA methylome landscapes of human primordial germ cells. | https://www.ncbi.nlm.nih.gov/geo/query/acc.cgi?acc=GSE63818 | Publicly available at the NCBI Gene Expression Omnibus (accession no: GSE63818) |
| Fagerberg L, Hall-ström BM, Oksvold P, Kampf C, Djur-einovic D, Odeberg J, Habuka M, Tah-masebpoor S, Da-nielsson A, Edlund K, et al | 2014 | Analysis of the human tissue-specific expression by genome-wide integration of transcriptomics and antibody-based proteomics | https://www.ebi.ac.uk/ar-rayexpress/experiments/E-MTAB-1733/ | Publicly available at ArrayExpress (accession no. E-MTAB-1733) |
| Lesch BJ, Silber SJ, McCarrey JR, Page DC | 2016 | Parallel evolution of male germline epigenetic poising and somatic development in animals. | https://www.ncbi.nlm.nih.gov/geo/query/acc.cgi?acc=GSE68507 | Publicly available at the NCBI Gene Expression Omnibus (accession no: GSE68507) |
| Yates A, Akanni W, Amode MR, Barrell D, Billis K, Carval-ho-Silva D, Cum-mins C, Clapham P, 650 Fitzgerald S, Gil L, et al | 2007 | A second generation human haplotype map of over 3.1 million SNPs. | ftp://ftp.ncbi.nlm.nih.gov/hapmap/recombina-tion/ | Publicly available at the NCBI ftp site for HapMap |
| Bherer C, Campbell CL, Auton A | 2017 | Refined genetic maps reveal sexual dimorphism in human meiotic recombination | https://github.com/cbherer/Bherer_etal_SexualDimorphismRecombi-nation | Publicly available at Github (https://github.com/) |

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
