## [Decision Letter]

Thank you for submitting your article "Biased gene conversion drives codon usage in human and precludes selection on translation efficiency" for consideration by *eLife*, and apologies for the delay. Your article has been reviewed by two peer reviewers, and the evaluation has been overseen by a Reviewing Editor and Patricia Wittkopp as the Senior Editor. The reviewers have opted to remain anonymous.

The reviewers have discussed the reviews with one another and the Reviewing Editor has drafted this decision to help you prepare a revised submission.

As you will see from the specific comments below, both reviewers agree that the analysis is convincing and ultimately worth publishing in *eLife*. However, they feel, and the reviewing editor agrees, that the results would be of more general interest if framed less as a response to a specific paper and more against the background of a long standing argument about the role of mutation and selection in shaping codon bias.

The two reviewers also made a number of specific suggestions that we would like you to address in revising your manuscript.

Finally, while you report an interesting association of recombination rates and expression levels, you present no evidence that the two are causally linked. Notably, the relationship could be mediated by histone marks, such as H3K4me3, associated with both recombination and expression, and it would be interesting to understand how H3K4me3 and PRDM9 binding sites mediate the observed effects. It is also unclear whether the idea of recombination interfering with transcription is plausible. In yeast and in birds, there is a weak *positive* association between meiotic expression levels and recombination. In mice, in turn, there is almost no recombination at promoters in meiosis (see Brick et al., 2012). In general, there are a number of papers on the determinants of recombination in human meiosis that may be relevant to this discussion (e.g., Pratto et al., 2014, as well as work by Bernard de Massy and Scott Keeney). We would therefore ask that you revise the text (and Abstract) accordingly.

*Reviewer #1 (major comments):*

For many years it has been debated whether codon usage bias in human genes reflects natural selection or non-selective evolutionary processes such as mutation rates or biased gene conversion. Recently it was proposed that differences in codon usage bias between different functional categories of genes is evidence for selection for optimal codon usage and translational efficiency (Ginghold et al., 2014). Specifically Gingold et al. found that genes in GO categories related to "proliferation" have a much different codon usage than genes in GO categories related to "differentiation". In this manuscript by Pouyet et al., the authors test whether differences in codon usage between functional gene categories can be explained by GC-biased biased gene conversion (gBGC) rather than natural selection.

The authors make several strong arguments that gBGC is a much better explanation for the observed differences in codon usage bias between different genes than natural selection. They find that codon usage bias described by Ginghold et al. (PC1 of a PCA) is almost perfectly correlated with the GC content of 3rd codon positions (GC3). GC3 is in turn very well predicted by a combination of intronic GC content, flanking GC content, recombination rate and meiotic gene expression. After controlling for these variables, the functional gene category explains very little of the variation in GC3.

Overall the paper is very clearly written and makes a convincing case that differences in synonymous codon usage between different GO categories is driven by gBGC. This result is not especially surprising given previous work showing that GC3 is well-correlated with regional GC content (isochores), but given the recent high-profile argument for selection by Ginghold et al. I feel that it is important to publish this finding. In addition, the paper is novel in that it proposes a mechanistic explanation for differences in GC3 between gene categories-that meiotic gene expression suppresses recombination so that genes with high meiotic gene expression undergo less gBGC and have lower GC content.

Comments:

The per-gene GC3 variance explained by meiotic expression is modest (R^2=8.3%) compared to that of intronic or flanking GC content (62% and 48%). If meiotic expression and reduction in rec. rate explain GC3 and variation codon usage, why is the correlation with meiotic expression so much weaker than the correlations with GCi and GCflank? It would be useful to include some acknowledgement and discussion of this in the paper. As shown in Figure 5, the correlation with meiotic expression and GC3 is far stronger at the level of gene categories. Is the explanation for the low R^2 for individual genes that individual gene estimates of meiotic expression are noisy? Or could it be that meiotic gene expression of broad gene categories has remained fairly consistent during evolution, even though the expression of individual genes has changed substantially?

The difference in R^2 between panels A and D of Figure 5 is puzzling. Why is the correlation between rec rate and GC3 so much stronger than the correlation between rec. rate and GCi? I would expect estimates of GCi to be more precise than those of GC3, since more sites can be used in the estimate, so differences in noise is not a good explanation. Is the better correlation with GC3 driven by the first exon (does the strength of correlation vary with distance from the promoter)? If so, this might suggest something about mechanism. E.g. gBGC might be highest near the promoter.

In the Abstract the authors say that meiotic transcription interferes with the formation of crossovers. While this might be true, the mechanism is uncertain and speculative. It would be better to draw a less speculative conclusion like "genes with higher meiotic transcription have lower recombination rates".

*Reviewer #2 (major comments):*

This article provides strong evidence that gene expression level during meiosis determines which synonymous codons are most likely to appear in human gene sequences. Compared to genes that are not expressed during meiosis, housekeeping genes that are highly expressed during meiosis are less likely to recombine, undergo GC-biased gene conversion, or have high GC content at synonymous sites. This conclusion is well supported by the analyses presented in the paper, which refute a claim by Gingold, et al. that a difference in human synonymous codon usage between "proliferation-related" and "differentiation-related" genes is driven by selection for translation efficiency.

The main weakness of this paper, in my view, is that it reads more like a response to Gingold, et al. than a standalone piece of work. To avoid the impression that the paper is a niche product that will only interest readers who have some kind of prior stake in the Gingold, et al. results, it would be helpful for the authors to convey a better sense of how Gingold, et al. sits in the broader landscape of selectionist explanations for codon bias, and what these new results mean for that work in general. In showing that selection on translation efficiency does not drive the contrast between codon bias in proliferation genes versus differentiation genes, are the authors only refuting the hypothesis of one particular paper, or of a broader set of papers claiming that selection for translation efficiency drives codon bias in the human genome?

Along the same lines, the title statement that "biased gene conversion drives codon usage" strikes me as underselling the results a bit. It doesn't give any hints about the intriguing and surprising observation that intron GC content and meiotic gene expression explain codon distribution so much better than isochore structure does. Once these results start being discussed in detail, the paper starts seeming less like a contradictory results response paper and more like a very interesting standalone paper, but this transition happens quite late in the manuscript.

---

## [Author Response]

*As you will see from the specific comments below, both reviewers agree that the analysis is convincing and ultimately worth publishing in* eLife*. However, they feel, and the reviewing editor agrees, that the results would be of more general interest if framed less as a response to a specific paper and more against the background of a long standing argument about the role of mutation and selection in shaping codon bias.*

We rewrote the Introduction, modified the title and the Abstract to present the study in the broader context of the ongoing debate on the role of selection in shaping codon usage in mammals.

*The two reviewers also made a number of specific suggestions that we would like you to address in revising your manuscript.*

*Finally, while you report an interesting association of recombination rates and expression levels, you present no evidence that the two are causally linked. Notably, the relationship could be mediated by histone marks, such as H3K4me3, associated with both recombination and expression, and it would be interesting to understand how H3K4me3 and PRDM9 binding sites mediate the observed effects. It is also unclear whether the idea of recombination interfering with transcription is plausible. In yeast and in birds, there is a weak* positive *association between meiotic expression levels and recombination.*

We agree: in species that lack Prdm9 (such as dogs, birds, arabidopsis or yeast), recombination hotspots are strongly enriched in active promoters (Auton et al., 2013; Choi et al., 2013; Singhal et al., 2015; Lam and Keeney 2015), which indicates that there is no mechanistic incompatibility between recombination and transcription activity in meiotic cells (this point is now mentioned in the Discussion).

*In mice, in turn, there is almost no recombination at promoters in meiosis (see Brick et al., 2012). In general, there are a number of papers on the determinants of recombination in human meiosis that may be relevant to this discussion (e.g., Pratto et al., 2014, as well as work by Bernard de Massy and Scott Keeney). We would therefore ask that you revise the text (and Abstract) accordingly.*

We added a section in the Discussion to present in more details the current knowledge about the determinants of the location of recombination events (and notably PRDM9) and to discuss the possible links with transcription (subsection “Relationship between meiotic expression and recombination”). Furthermore, we reevaluated the relationship between intragenic recombination and expression using DSB maps (Pratto et al., 2014) and a new high resolution genetic map that has just been published (Bhérer et al., 2017) (Results subsection “The difference in intragenic recombination rate between functional categories is explained by their expression level in meiosis”, Figure 4, Figure 3—figure supplement 3; Figure 4; Figure 4—figure supplement 1 and Figure 4—figure supplement 2). We have modified the text, the figures and their legends to specify which datasets were used to estimate recombination (intragenic crossover rates, or DSB hotspots). We updated the supplementary information (http://doi.org/10.5281/zenodo.835063) to include these new data and analyses.

These new data show that the relationship between crossover rate and meiotic gene expression is even stronger than initially reported: we observed that the crossover rate is 3.5 (males) to 5.4 (females) times lower in highly expressed genes (top 10%) compared to weakly expressed genes (bottom 10%) (Figure 3, Figure 3—figure supplement 3). This decrease affects the entire transcription unit, from the TSS to the polyadenylation site (Figure 4). The reduction in crossover rate is due, at least in part, to a lower density of meiotic DSB hotspots within highly expressed genes (Figure 3—figure supplement 3; Figure 4—figure supplement 2). One possible explanation for this pattern, is that high transcription levels might interfere with the binding of PRDM9 (or with its histone H3K4 trimethylation activity), and thereby decrease the rate of DSB formation within the transcription unit (these results are discussed in more details in subsection “Relationship between meiotic expression and recombination”). We agree that these observations are not sufficient to demonstrate a causal link between meiotic expression and recombination rate. We corrected all sentences (Abstract, main text) that were suggesting a direct impact of transcription on recombination).

Reviewer #1 (major comments):

*[…] The per-gene GC3 variance explained by meiotic expression is modest (R^2=8.3%) compared to that of intronic or flanking GC content (62% and 48%). If meiotic expression and reduction in rec. rate explain GC3 and variation codon usage, why is the correlation with meiotic expression so much weaker than the correlations with GCi and GCflank? It would be useful to include some acknowledgement and discussion of this in the paper. As shown in Figure 5, the correlation with meiotic expression and GC3 is far stronger at the level of gene categories. Is the explanation for the low R^2 for individual genes that individual gene estimates of meiotic expression are noisy? Or could it be that meiotic gene expression of broad gene categories has remained fairly consistent during evolution, even though the expression of individual genes has changed substantially?*

The referee is correct: the SCU of individual genes depends primarily on the isochore in which they are located (i.e. large-scale long-term variation in recombination rate), and only secondarily on their meiotic expression level (which affects locally the intragenic recombination rate) (Table 1). At the level of gene categories, meiotic expression explains a high fraction of variation in SCU. This is due to the fact that in gene set analyses, SCU is averaged over a large number of genes, located in different isochores, which leads to decrease the isochore effect among functional categories (and hence mechanically increase the fraction of the variance explained by expression).

We have added a paragraph in Discussion (subsection “Biased gene conversion drives codon usage in human” paragraph five) to clarify this point.

That said, the correlation with meiotic expression levels is probably blurred by the variance caused by measurement errors and temporal fluctuations, as suggested by the reviewer. In the main text, we have been using the average of 4 samples for male meiotic expression (from Lesh et al., 2016). We verified that the correlation between GC3 and meiotic expression level is always lower (by 5%-25%) when expression level is measured using a single sample. Hence, measurement errors influence the level of the correlations. We also verified that expression level at male meiosis is well correlated between mouse and human (R²=0.73, N=14,768 genes with 1:1 orthologs in mouse and humans; data from Lesh et al., 2016). Hence, expression levels measured in humans are good predictors of human/mouse average meiotic expression level, and thereby carry information on long-term intragenic recombination. Yet expression levels have changed for many genes between human and mouse. We measured the correlation between GC3 and expression level on a set 9185 genes for which expression level is roughly comparable in human and mouse (assessed by the fact that they rank in the same quartile in both species). The correlation measured on this set of genes is increased by 20% as compared to the correlation measured on the total set of genes. Hence, the strength of the correlation between GC3 and meiotic expression level reported in the paper is probably a lower bound because of these two sources of variability. In any case, the local effect of meiotic expression is much weaker than the large-scale isochore effect.

*The difference in R^2 between panels A and D of Figure 5 is puzzling. Why is the correlation between rec rate and GC3 so much stronger than the correlation between rec. rate and GCi? I would expect estimates of GCi to be more precise than those of GC3, since more sites can be used in the estimate, so differences in noise is not a good explanation.*

At neutral sites, the base composition of sequences is expected to depend on the pattern of base substitution (i.e. on the pattern of base-replacement mutations and on the intensity of gBGC), and on the pattern of deletions and insertions. The difference between GCi and GC3 can be explained by the fact that while the base composition of exons is almost exclusively driven by base substitutions, introns are also affected by deletions and insertions. Thus the base composition of introns depends not only on the long-term intensity of gBGC, but also on the pattern of indels (notably of transposable elements) (Duret and Hurst 2001 Mol. Biol. Evol. 18:757–62). Moreover, there is evidence that recombination rates differ between exons and introns (Kong et al., 2010 Nature 467:1099–103).

*Is the better correlation with GC3 driven by the first exon (does the strength of correlation vary with distance from the promoter)? If so, this might suggest something about mechanism. E.g. gBGC might be highest near the promoter.*

Indeed, Bhérer et al., 2017 showed that there is an elevation of recombination rate around the TSS, and hence gBGC is expected to be stronger near the promoter. However, the variation in intragenic recombination rate according to meiotic expression affects the entire transcription unit (Figure 4). Hence, the variation in recombination rate (and hence SCU) among functional categories is not expected to be restricted to the 5’ end of genes.

To verify this point, we computed the correlation between recombination rate and GC3 separately for the first or the last parts of the genes. For simplicity, we extracted 50 codons at the beginning and at the end of each gene (50 codons is the average length of human coding exons (Lander et al., 2001)), and we computed the average GC3 per GO category. The correlation between GC3 and recombination rate is lower for the beginning (R2=0.28; see Author response image 1) than for the end of the coding sequences (R2=0.37). Thus, the correlation between recombination rates and GC3 is not driven by the codons proximal to the promoter.

The new results about variation in recombination rates along genes according to their expression level are now presented in the revised manuscript (Figure 4). We therefore think that the supplementary analysis presented here (Author response image 1) is not necessary. If you consider that it is important, we would be ready to include it as supplementary material.

**Author response image 1. respfig1:** Correlation between between intragenic crossover rate and GC3 (measured in the first or last 50 codons of genes), among functional gene categories.

*In the Abstract the authors say that meiotic transcription interferes with the formation of crossovers. While this might be true, the mechanism is uncertain and speculative. It would be better to draw a less speculative conclusion like "genes with higher meiotic transcription have lower recombination rates".*

The reason why intragenic recombination rate correlates negatively with meiotic expression level is not known. We have modified the Abstract and the main text in line with the suggestion of the reviewers and the editor. We have also introduced a discussion of the link between meiotic transcription and crossovers in the main text (subsection “Relationship between meiotic expression and recominbation).

Reviewer #2 (major comments):

*[…] The main weakness of this paper, in my view, is that it reads more like a response to Gingold, et al. than a standalone piece of work. To avoid the impression that the paper is a niche product that will only interest readers who have some kind of prior stake in the Gingold, et al. results, it would be helpful for the authors to convey a better sense of how Gingold, et al. sits in the broader landscape of selectionist explanations for codon bias, and what these new results mean for that work in general. In showing that selection on translation efficiency does not drive the contrast between codon bias in proliferation genes versus differentiation genes, are the authors only refuting the hypothesis of one particular paper, or of a broader set of papers claiming that selection for translation efficiency drives codon bias in the human genome?*

*Along the same lines, the title statement that "biased gene conversion drives codon usage" strikes me as underselling the results a bit. It doesn't give any hints about the intriguing and surprising observation that intron GC content and meiotic gene expression explain codon distribution so much better than isochore structure does. Once these results start being discussed in detail, the paper starts seeming less like a contradictory results response paper and more like a very interesting standalone paper, but this transition happens quite late in the manuscript.*

We rewrote the Introduction and modified the title and the Abstract to present the study in the broader context of the ongoing debate on the role of selection in shaping codon usage in mammals.